# Telomeres and Their Neighbors

**DOI:** 10.3390/genes13091663

**Published:** 2022-09-16

**Authors:** Leon P. Jenner, Vratislav Peska, Jana Fulnečková, Eva Sýkorová

**Affiliations:** Institute of Biophysics of the Czech Academy of Sciences, CZ-61265 Brno, Czech Republic

**Keywords:** satellite, telomere evolution, interstitial telomere sequences, retroelements, subtelomere structure, telomerase RNA, TRAP, FISH, NGS, eukaryotic tree of life

## Abstract

Telomeres are essential structures formed from satellite DNA repeats at the ends of chromosomes in most eukaryotes. Satellite DNA repeat sequences are useful markers for karyotyping, but have a more enigmatic role in the eukaryotic cell. Much work has been done to investigate the structure and arrangement of repetitive DNA elements in classical models with implications for species evolution. Still more is needed until there is a complete picture of the biological function of DNA satellite sequences, particularly when considering non-model organisms. Celebrating Gregor Mendel’s anniversary by going to the roots, this review is designed to inspire and aid new research into telomeres and satellites with a particular focus on non-model organisms and accessible experimental and in silico methods that do not require specialized equipment or expensive materials. We describe how to identify telomere (and satellite) repeats giving many examples of published (and some unpublished) data from these techniques to illustrate the principles behind the experiments. We also present advice on how to perform and analyse such experiments, including details of common pitfalls. Our examples are a selection of recent developments and underexplored areas of research from the past. As a nod to Mendel’s early work, we use many examples from plants and insects, especially as much recent work has expanded beyond the human and yeast models traditional in telomere research. We give a general introduction to the accepted knowledge of telomere and satellite systems and include references to specialized reviews for the interested reader.

## 1. Introduction

The essential DNA structures that form eukaryotic chromosomes are centromeres, telomeres and origins of replication. Centromeres are vital for proper nuclear division and telomeres protect the ends of linear chromosomes from attrition during DNA replication. Each chromosome also possesses genes which populate the chromatin regions lying between centromeres and telomeres. Genes are sequence-based quanta of information, Gregor Mendel’s “elements”, that are the foundation of organism identity when realized. A simplified picture of genetics was initially recognized by Mendel without any deep knowledge of DNA, RNA and proteins; molecules that we now know realize organism function. When thinking about Mendel 200 years after his birth, it is obvious that simple methods and an open mind are the solution to many scientific questions, even those related to gene function, genome structure and evolution. Of course, there is also a little bit of luck in choosing a model. Mendel’s choice, the garden pea (*Pisum sativum*) has a large genome that has only recently been sequenced [1] and is not a popular organism for modern plant genetics due to the abundance of repetitive elements in its genome. Conversely, another one of Mendel’s favorite organisms, the honeybee (*Apis melifera*, [2]) is still intensively studied as it offers a chance to understand the phenomena of social life and cooperation in the insect world. It is an extraordinary coincidence that the dawn of telomere biology in 1938 is linked to maize and *Drosophila*, models which were used in the pioneering works of McClintock and Muller, who showed breakage-fusion-bridge cycles and chromosome healing after X-ray damage [3,4]. At that time, it was assumed that the entire chromosome was filled with genes, and Muller’s original definition of the telomere speaks about it as a special terminal gene. Today, the definition of a gene has changed and we know more about chromosomes and telomeres, but there is still much yet to be discovered (e.g., [5,6,7,8,9,10,11,12,13] and references herein). Telomeres and satellites are often neighbors at chromosome termini. Much telomere research derives from human, ciliate and yeast models. Knowledge of telomere evolution has increased enormously in recent years, however, thanks to discoveries in plant and insect models. Here we tell a story of (mostly) plant telomere and subtelomere research in which principles and experimental approaches are illustrated to guide researchers interested in species diversity, genome structure and evolution. We will describe the methods which have been used, namely those that represent a good standard in this area of research, and are widely accessible to the scientific community. We also highlight especially interesting areas where researchers are still exploring. We will refer to reviews and original papers that we recommend for further reading and which will be useful background for the critical evaluation of experimental and in silico results.

## 2. Genomes Are Mostly Repeats 

Pioneering work on the purification of eukaryotic DNA used cesium chloride gradients to separate DNA by density. During these experiments, distinctive satellite DNA bands (Figure 1a) were observed, with different buoyant density to the major genomic fraction [14,15,16,17,18] These DNA fractions were found to be comprised of various repetitive sequences and their separation is due to the physical properties that arise from their different base pair compositions and GC content. The majority of repeats are organized in long tandem arrays in a head-to-tail fashion. Cytologically, clusters of satellite DNA were found in various locations in the chromosomes, frequently in centromeric, pericentromeric and subtelomeric regions. These represent a remarkably large part of the genomes of diverse eukaryotes including *Drosophila*, mice and plants [1,19,20,21,22,23,24,25,26,27]. There are further classes of repetitive sequences which also make up a proportion of the genome similar to or larger than satellite DNA. In contrast to satellites, these are more often dispersed throughout the genome rather than clustered in one or a few locations, e.g., mobile elements (e.g., [5] and references herein). In *sensu stricto*, the name satellite DNA refers to those various repeats found in huge clusters of different buoyant density, but the term is used generally for any tandem repeat. A simple classification of tandem repetitive sequences is based on the length of both the monomer repeat unit and the overall repeated sequence, microsatellites (usually <5 bp monomers, <150 bp total), minisatellites (<25 bp, <20 kb total) and satellites (>150 bp, up to Mb size). However, these size categories are somewhat arbitrary, and the same terms are applied differently in the work of different researchers [28,29].

Examining the structure of the chromosome, it is apparent that the majority of telomeric sequences are minisatellite repeats (see [7,11,30] for review). In human centromeres, large arrays of repeated α-satellite DNA consist of tandem, head-to-tail repeats of a 171-bp monomer that are further organized into higher order repeats (see [8] for review). Satellite repeats form the centromeres of many organisms (see [6,31] for review, examples in Figure 2). In contrast to telomeric sequences (see below), the centromere satellites of even closely related species are often unrelated in sequence, and the potential for satellites to form centromeres cannot be predicted, only experimentally proven [25]. Ribosomal DNA (rDNA) units are not usually classified as satellites, but they also cluster as tandem repeats in tens to thousands of copies in genomes (see details and databases in [32,33]). A functional link between satellite repeats and human diseases was found, e.g., for microsatellites, however, the majority of reports focus on the importance of repetitive sequences for chromatin composition, discussed in specialized reviews (see [34,35,36,37,38]).
Figure 1Summary of experimental and in silico approaches leading to candidate satellite identification (**a**–**h**) and examples of satellite characterization (**i**–**m**). Classical experimental approaches (**a**–**c**) rely on basic physical principles. (**a**) Satellite DNA bands of different buoyant density are separated from the main gDNA by CsCl gradient centrifugation. (**b**) Single stranded DNA fragments are reassociated for a set time at high temperature to form C_0_t fractions. High-copy sequences bind to complementary DNA strands in short time intervals. After removal of unassociated ssDNA strands by S1 nuclease and/or hydroxylapatite chromatography, C_0_t fractions can be used for localization by FISH and/or for library construction. (**c**) Tandem repeat sequences can contain conserved sites for restriction endonucleases. After restriction digestion of gDNA, high-copy satellites form distinct bands visible on agarose gels that can be cut out and cloned. (**d**) gDNA samples can be used directly for Next Generation Sequencing (NGS) or Third Generation Sequencing (TGS) and data further processed in silico [39,40], e.g., using RepeatExplorer. Approaches (**e**,**f**) need state-of-the-art technologies and specific sample preparation. (**e**) When specific morphology parameters of chromosomes can be determined, partial or whole chromosomes can be microdissected and amplified by degenerated-oligonucleotide-primed (DOP) PCR, e.g., terminal fragment library construction from the X chromosome of dioecious plant *Silene latifolia* [41] (see also (**l**)). (**f**) Differences between the size of sex chromosomes and autosomes of *Rumex acetosa* enabled separation of chromosome-specific samples by flow cytometry [42]. (**g**) Vectorette PCR [43] is a genome-walking approach that enables the amplification of specific DNA fragments in situations where the sequence of only one primer is known. A vectorette-like strategy can be used to identify sequences associated with the telomere (TAS, telomere associated sequences, red) and with internal telomeric sequences (ITS, magenta). After restriction digestion of gDNA, fragments are ligated to the vector and PCR reaction with the vector-specific (black) and telomeric C-rich (blue) primer will produce a mixture of TAS and ITS sequences, e.g., identification of the subtelomeric sequence from *Nicotiana tomentosiformis* [44]. (**h**) Telomere-subtelomere junction PCR can be employed to demonstrate telomere attachment when a candidate subtelomeric sequence is known, e.g., to rDNA in *Arabidopsis* [45], the subtelomeric HRS60 satellite in tobacco [46] or the subtelomeric X43.1 sequence from *S. latifolia* [47,48]. The latter identified direct attachment of X43.1 to the telomere or via a linker formed by another satellite sequence 15Ssp (green) (see also results on (**l**,**m**)) or by various low-copy linker sequences (grey). (**i**) Libraries created by the approaches described above can be searched for high-copy repetitive sequences by colony hybridization using labeled (*) gDNA or C_0_t fraction [49,50] as a probe. Candidate clones can be used directly for localization by FISH, characterization of the genomic arrangement by Southern hybridization (examples in (**l**)) and/or sequenced. After sequencing, sequence-specific primers can be designed for PCR amplification of candidate sequence from gDNA (see (**k**)). Examples: (**j**) The ACSAT repeat was isolated as in (**a**) from *Allium cepa* gDNA [14] and localized by FISH (yellow) to subtelomeres on metaphase chromosomes of *A. cepa* (left panel). Only chromosomes of *A. cepa* showed ACSAT-specific signals on metaphase of an interspecific hybrid between *A. cepa* and *A. kurssanovii* (right panel) [51]. (**k**) Typical ladder of PCR products with periodicity corresponding to repeat unit length, demonstrating tandem organization of ACSAT, 15Ssp and X43.1 repeats when amplified with sequence-specific primers from 1 ng or 0.1 ng of gDNA of *A. cepa* and *S. latifolia*. (**l**) Southern hybridization demonstrated tandem organization of 15Ssp and X43.1 satellites in contrast to the dispersed pattern of X12, X36 and X41 repetitive sequences that were previously localized in subtelomeres of *S. latifolia* [41]. (**m**) FISH with LNA (locked nucleic acid) oligonucleotide probes of the X43.1 and 15Ssp satellites (upper panel, green and red, respectively) shows various localization and co-localization signals on metaphase chromosomes of *S. latifolia* [52]. The CL172 satellite (green, lower panel) is detected on chromosomes of *A. sativum* [53]. Pictures were adapted by courtesy of Prof. Ingo Schubert ((**j**), [51]), Dr. Terezie Mandáková ((**m**), [53]) and Dr. Eduard Kejnovský ((**m**), [52]); scale bars, 10 µm; chromosomes are counterstained with propidium iodide (**j**) or DAPI (**m**). Data in (**c**,**k**,**l**) are from Sýkorová (unpublished and [48]).
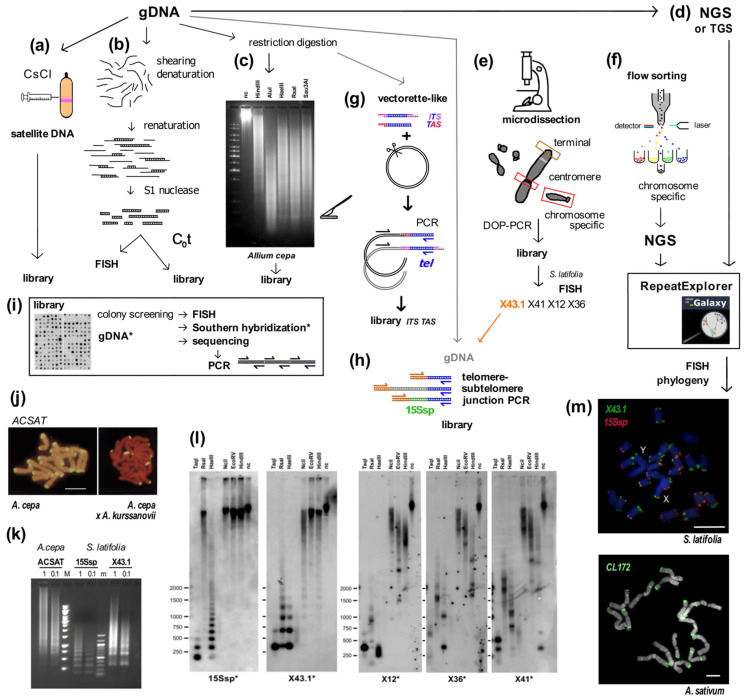



Eukaryotic genomes are full of repetitive sequences with no obvious specific function and such sequences increase genome size without increasing gene number. For these reasons repetitive sequences were often referred to as a “junk” or “selfish” DNA [54]. However, after decades of satellite research, data suggest that these regions can be transcribed and participate in genetic and epigenetic regulation [55,56,57,58]. Repetitive sequences seem to be involved in bulk chromatin organization, chromosome pairing and genome stability, and might protect genes from damage simply by increasing the probability that random environmental damage will occur in “junk” regions (e.g., [36,59,60,61], see [31,62] for review). Conversely, small genomes exist happily besides large genomes [28,63] so repeats or even chromosomes can clearly be lost, reducing genome size without loss of function. For example, satellite, rDNA and other repeats are major components of B chromosomes (supernumerary chromosomes) which occur in approximately 15% of eukaryotes [13,64,65]. B chromosomes are known to be dispensable for normal development, do not recombine with standard A chromosomes and exhibit non-Mendelian inheritance. As a result, their number varies between individuals with no obvious repercussions (e.g., *Zea mays*, 2n = 20 As + 0–34 Bs (see [13] and references herein). Equally, it is a well-known fact that many agronomically important crops are polyploids and/or have larger genomes and plenty of repetitive sequences, so clearly there must be some benefit to this additional DNA (e.g., [1,13,34,66]). It is noteworthy that there are thousands of understudied crop species across the globe in which satellite research could be beneficial to describe genome organization for breeders and to better understand their evolution.

Mechanisms which maintain repetitive sequences and pathways, deciding which repeats will be kept and which will be lost during genome instability, are not well understood [6,35]. Nevertheless, considerable research has gone into characterizing concerted evolution, the rapid spread and stabilization of mutations within a population, which often occurs disproportionately in or near repetitive sequences ([67], reviewed in, e.g., [6,36,62]). The progress of such mutations across related individuals is described as “molecular drive” and can be classified into one of six transition states depending on the extent to which a mutation is present in an individual in comparison to a reference genomic sequence [68]. Although much progress has been made in explaining this complex area, many molecular mechanisms for satellite mutation still await experimental examination (reviewed in [69]). It is known that unequal crossover events between paired chromosomes during meiosis are likely to be responsible for gene duplication of the α-chain of hemoglobin in primates; this gene is adjacent to repetitive sequences which are theorized to be prone to misalignment during DNA recombination, leading to the duplication or deletion of genes [67]. The more complex variations of the S1 pericentromeric repetitive sequences across related European brown frog species feature large expansions or contractions of repeat arrays. Examination of frog data led to the proposal that new arrays can arise through the loss of sections of repeats through intramolecular homologous recombination followed by rolling circle replication of repeated arrays [70]. Overall, these mechanisms and others are thought to give rise to an overall “library“ of repetitive sequences that are generally conserved or, at most, only change slowly across a species, even if they are locally gained or lost in individuals [71].

## 3. Fellowship of the Satellites

Interestingly, the TTAGGG repeat was first reported as the α-satellite of guinea pig and kangaroo rat in the 1970s [72,73], (see [21] for review). Today we recognize this repeat as the typical telomere sequence, not only of vertebrates [74] but of many other organisms (see [7,30,75] for review). The majority of identified telomere repeats are of minisatellite size and maintained by a special enzyme, telomerase ([76,77,78], reviewed in, e.g., [7,11,79]). There are also several well-known examples of non-telomerase alternatives from Diptera (e.g., *Chironomus*, *Anopheles*, *Drosophila* [80,81,82,83,84]), which we describe briefly here. Telomeres of *Chironomus* species are formed by ca. 350-bp-long satellite repeats and are maintained by homologous recombination as demonstrated in *Anopheles*. This recombination-based telomere maintenance exists in all eukaryotic cells as a backup mechanism which is employed in situations where telomerase is lost, e.g., in human ALT cells, yeast Type II survivors or knockout plant mutants [85,86,87,88], reviewed in [75]. Among insects, a special telomere is formed by a cluster of retroelements, a phenomenon that was first identified in *Drosophila* [89,90,91,92,93,94], together with satellite and other repeats representing 60% of the total genome (see [11,35] for review). Research of these high-copy sequences has proven important for the general understanding of chromatin biology not only for this model and is covered in detail in many reviews [95,96,97]. The evolution of a telomere maintenance mechanism that uses telomere-targeting transposable elements instead of telomerase, present exclusively in Diptera, is discussed in reviews [11,75,98,99].

The subtelomere is often referred to as a buffer zone between the telomere and the internal chromosome and is a hot-spot for recombination. This is advantageous for sequence variations of the *VSG* (variant surface glycoprotein) gene families localized exclusively in subtelomeres of parasites (reviewed in [100]). However, in most organisms, the subtelomere is formed of repetitive sequences including satellites and rDNA that could be attached to telomeres either directly or by specific linker sequences (Figure 1g,h). Due to their tandem repetitive nature and often high copy number, satellite sequences are popular probes used for karyotyping and taxonomy. For example, the 184-bp-long HRS60 repeat sequence [101] serves as a specific probe for the S-genome in the allotetraploid *Nicotiana tabacum* and other polyploids originating from the *Nicotiana sylvestris* parental diploid [102,103]. The 50-bp-long PisTR-B repeat shows mostly subtelomeric and pericentromeric signals whose patterns, together with chromosome morphology, allow the discrimination of all chromosome types within the garden pea karyotype [50,104]. Another example is the 375-bp-long repeat ACSAT that is specific to closely related *Allium* species from the section Cepa but is not present in other *Allium* species [14,51,105] (Figure 1j). ACSAT satellites form 4% of the *A. cepa* genome (1C = 17.9 pg, [106]) and are located in subtelomeric regions. Similarly, rDNA repeats are found in subtelomeric positions in many plant species, including model *Arabidopsis thaliana* [45,107]. Here, direct attachment of rDNA to telomeres enabled the study of changes in telomere length, rDNA copy number and chromatin composition at a molecular level in CAF-1 (chromatin assembly complex) mutants [108,109]. 

When repetitive sequences are characterized, the relationship between species, chromosomal arrangements and genome evolution can often be studied in detail [110,111]. The classic and still popular way to identify and characterize satellite repeat sequences experimentally starts with restriction digestion of genomic DNA, followed by isolation of restriction fragments from a gel and ligation to a vector (more examples of experimental and in silico approaches are summarized in Figure 1, e.g., [14,27,41,42,44,53,111]). Cloned sequences can be used as probes in Southern hybridization (Figure 1l) for characterization of repeat arrangements. Satellite sequences show a typical ladder of bands, the periodicity of which corresponds to the length of repetitive unit and suggests tandem repeat organization. Similar ladders of products would be observed when the repeat is amplified from genomic DNA using sequence-specific primers (Figure 1k). Cloned sequences or amplified PCR products can then be used for fluorescent in situ hybridization (FISH) experiments to determine the position of sequences in chromosomes (Figure 1m). There is a good chance that a novel satellite will be detected as a distinct signal representing a cluster of tandem repeats, in contrast to high-copy repetitive sequences that are not organized as tandem repeats and thus usually show a variety of scattered interstitial signals. Today, NGS and TGS data can be mined to retrieve candidate tandem or other repeats (Figure 1d,f) for experimental characterization and to reconstruct phylogeny ([40,112,113], see below). In two elegant examples, NGS mining studies were used to characterize dioecious plants using a comparison of male and female seabuckthorn [27], and NGS data generated from flow-sorted samples were used to uncover specific repetitive sequences in autosomal and sex chromosomes in *R. acetosa* (Figure 1f, [42]).

## 4. Telomeres as Steps in Species Evolution

To begin with, telomere DNA sequences were assigned as a trait of a large group of organisms, e.g., TTAGGG in vertebrates, TTTAGGG in plants, TTAGG in insects/arthropods [74,114,115,116,117] (see [75] for review). This conservation has proven advantageous in microscopy studies and telomeric probes are second only to rDNA probes [32,33,118], e.g., to distinguish and study telocentric chromosomes, to recognize Rabl-like or bouquet organization or various chromosomal aberrations [119,120,121,122,123,124,125,126]. Numerous reports that characterized typical telomeric sequences in an increasing number of species seemed to confirm the telomere consensus T_x_A_y_G_z_. Telomeric sequences in yeast models, e.g., TG_1–3_ in budding yeast [127,128], T_1–2_ACA_0–1_C_0–1_G_1–6_ in fission yeast [129], 8–25 bp-long repeats in *Kluyveromyces* and *Candida* [130,131] were treated as an interesting variety from the general repeat unit T_x_A_y_G_z_ and special only to yeast. Current research on Saccharomycotina [132,133] has revealed even more telomeric variants, although despite their considerable divergence, all of these telomere sequences have guanines (Gs) as one of their most conserved features [131,132,133,134].

Missing signals using telomere probes in in situ hybridization experiments were the first hints towards identifying organisms that do not possess typical telomeres formed by the expected repeat, e.g., plants *Allium* (Asparagales, [114]), *Cestrum* (Solanales, [135]), some beetles and the spider *Tegenaria ferruginea* [117]. In the next few years, detailed studies revealed gradually more species with unknown telomeres from plants [136,137] and insects [138]. This led to a breakthrough in the general view of telomeres. Studies that mapped telomere sequences in plants, animals and algae identified evolutionary switchpoints in which sequences typical to one group were replaced by other variants [30,135,139,140,141,142,143,144,145]. For example, a group of species from the plant order Asparagales changed their telomeric sequence from the *Arabidopsis*-type repeat TTTAGGG to the human-type TTAGGG. An elusive, highly divergent telomere repeat was finally identified in *Allium* (Amaryllidaceae, Alloidae, [146], see Section 7), one of the largest monocotyledonous genera with an estimated 800–900 species [147]. Similar step changes were found in green algae, in which the transitions from TTTAGGG to novel types TTAGGG, TTTTAGGG or TTTTAGG allowed the grouping of species with the same telomere in distinct phylogeny clades [30,142,148]. A similar switch was identified in beetles where the repeat TTAGG was replaced with the TCAGG repeat [143]. A broad experimental study of algal telomeres, accompanied with the identification of candidate telomeric sequences from genomic databases of various species across the eukaryotic tree of life, showed TTAGGG and TTTAGGG telomeres as being the predominant telomeric types [30]. Fulneckova and colleagues [30] mapped the occurrence of telomeric sequences in phylogeny revealing the TTAGGG repeat as an ancestral eukaryotic telomere and current phylogeny [149,150] still supports this hypothesis (see [7] for review). Interestingly, just as many telomere variants were experimentally verified, many more species and groups with unknown telomeres were discovered [11,30,142,148,151]. Telomere sequence variants and their evolution in plants and algae are described in detail in a review by Peska and Garcia [7]. Progress in insect telomere identification is reviewed in Mason et al. [11] and recent findings are mapped in [151,152,153,154].

## 5. Telomere Minisatellites Are Much like Any Other DNA Sequences

When exploring the occurrence of telomere minisatellite repeats in the genome, we should keep in mind that telomere-like sequences can occur in locations other than in the telomere. Such sequences are called interstitial telomeric sequences (ITSs) and can be classified as part of several groups according to their length, occurrence and structure (recently reviewed in [155], Figure 2). ITSs can have the same sequences as telomeres or they can have variant telomere-like repeats. For example, budding yeast has the telomeric sequence TG_1–3_ and interstitial tracts of TTAGGG repeats are present in subtelomeric and other regions [156]. ITSs can occur as a few copies across the genome, including regions that are proximal to genes, but also in clusters found frequently in pericentromeric or subtelomeric regions. The arrangement of ITS sites can also be classified in respect to the orientation and composition of telomere-like sequences as head-to-tail or head-to-head, homogeneous or degenerated tandem repeats and with or without linker sequence(s) (Figure 2b). When ITSs occur in a head-to-head orientation with a linker sequence, these can be amplified using a single-primer PCR reaction [157] (Figure 2c). ITSs can be unique or part of longer repetitive sequences and are a suitable genetic marker for mapping [157,158,159]. ITSs in clusters usually contain a large portion of degenerate telomeric motifs and could be interspersed with other repetitive sequences [158,159,160].
Figure 2**Experimental examination of ITS and telomeric repeats.** (**a**) Telomere repeats are strand oriented. (**b**) Telomere-like repeats in telomeres or internal sites may form clusters or short stretches. Single-primed PCR distinguishes between these using an extension reaction with a single telomeric oligonucleotide primer (C-rich primer is shown, triangles). Telomeric sequences, short and clustered ITSs produce a smear of ssDNA products visible after hybridization with a radioactively (*) labeled probe (right, e.g., from *Chlorela vulgaris*, experiment performed as in [135]). Cloneable dsDNA products visible in an ethidium-bromide stained agarose gel (etd) are produced when ITSs occur in head-to-head orientation. When dGTP is omitted, bands are not produced by ssDNA or short ITSs, but ssDNA from a telomere is elongated until primer extension stops at the first G in the subtelomere. This reaction showed the *Arabidopsis*- and human-type telomere repeats are absent in *Allium* and *Cestrum* [135,140,161]. (**c**) Different patterns of ITSs amplified from four *Cestrum* species in single-primed PCR using C-rich and G-rich primers for the *Arabidopsis*-type telomeric repeat [135,157]. (**d**) The specific pattern of ITS-associated sequence BR23 (green) was visualized on *Cestrum elegans* chromosomes using FISH. The high-copy repeat BR23 shows dispersed and clustered signals (5S rDNA in red, counterstained with DAPI; adapted from [157]). (**e**) Allotetraploid *Cardamine scutata*, a hybrid of *C. parviflora* and *C. amara* with the parental origin of chromosomes visualized by GISH (left panel, GISH) and telomeric probe (TEL) that detects differing pericentromeric ITS clusters (adapted from [159]; modified). (**f**) FISH of the 180-bp centromeric satellite (*CEN180*), retroelement *ATHILA* and *TEL* on pachytene chromosomes of *A. thaliana.* Interstitial telomeric locus in the pericentromeric region of chromosome Ch1 is marked by an arrow (adapted from [160]; modified). (**g**) TRF (terminal restriction fragment) method visualizes telomeric and ITS fragments from *A. thaliana* after restriction digestion of gDNA with *Mse*I. (**h**) Schema illustrating the effect of Bal31 nuclease digestion on telomeric, subtelomeric, ITS and internal genomic sequences. After DNA isolation, DNA is fragmented and Bal31 nuclease gradually shortens these fragments from the end. Bal31-digested samples can be used for specific telomere-subtelomere PCR (left, see below). Further restriction digestion (right, H) results in the visualization of TRF signal (h-tel) shortening and verification of the terminal position of a candidate sequence. (Left) PCR/qPCR investigation of genomes with short telomeres (e.g., *A. thaliana,* see results in (**i**–**k**) adapted from [162]) proving subtelomeric position of candidate sequences (A,B). When the telomere is completely digested, PCR with a C-rich primer cannot amplify the product (tel-a, tel-b), and further digestion results in a loss of amplification signal from subtelomere regions proximal to telomeres (A) in contrast to ITS (C, pericentromeric ITS in *A. thaliana*, see schemas in (**j**)) or control sequences (D,E). Bal31 nuclease also degrades ssDNA (F) and some dsDNA sites with altered structures (G). (**i**) Dynamics of Bal31 digestion monitored by qPCR. Short gDNA exposure to Bal31 results in a sudden, seemingly non-specific decrease in gDNA amount followed by a gradual decrease over a prolonged time. (**j**) Bal31-sensitivity of specific subtelomeric sequences from chromosome arm 2R (pat and gal2) and the resistance of the centromeric ITS region to Bal31 digestion resolved by PCR. gDNA integrity was monitored by amplification of 5 kb-long fragments of the *TERT* gene. (**k**) qPCR analysis of specific subtelomere (gal2, pat, gal5), ITS and control sequences documented a decrease of subtelomeric sequences in relation to their position in the subtelomere. Relative DNA levels were calculated by the ΔCt method (**i**) or ΔΔCt method [163] using ubiquitine-10 as a reference gene relative to the nontreated DNA sample (**k**). Color coding is the same for (**h**–**k**). Pictures were adapted by courtesy of Dr. Terezie Mandáková (**e**,**f**) and Prof. Andrew Leitch (**d**), scale bars are 10 µm.
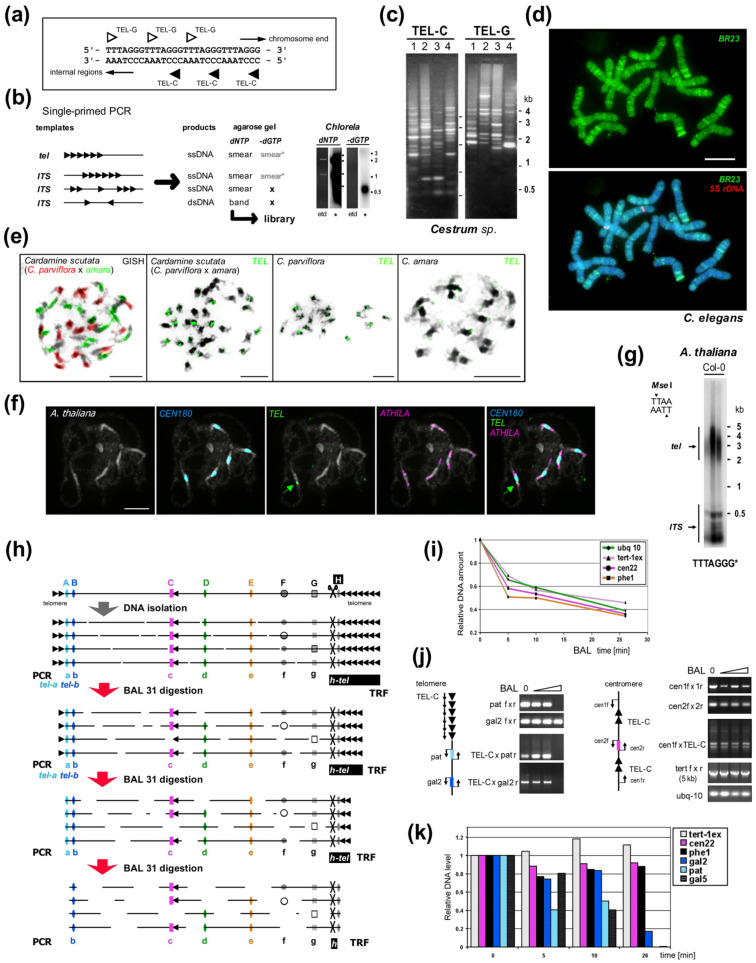



When such clusters are big enough, these can be detected by FISH (Figure 2d–f) and distinguished from telomeres (e.g., [49,157,159,164,165,166,167,168,169]). If they are shorter than the detection limit of this method, they can still show a positive signal when investigated by Southern hybridization or primer extension (Figure 2g). The origin, evolution and function of ITSs are still subject to much discussion [120,155,169,170,171,172,173]. The massive areas of ITSs often found in pericentromeric regions can be explained as the result of mechanisms such as unequal gene conversion, crossing-over, DNA replication slippage and rolling circle replication of extrachromosomal circular DNA. Some ITSs co-localize with sites of chromosomal breakage and are described as remnants of ancient chromosomal rearrangements, such as during primate evolution [174]. A similar view holds for human ITSs arranged as head-to-head blocks of telomeric repeats that seem to result from the terminal fusion of ancestor chromosomes [126,175].

We are still far from understanding the interplay of mechanisms that are activated during genome instability. It has long been considered that overall change in chromosome architecture can result from breakage-fusion-bridge cycles, a phenomenon first described in maize ([176], reviewed in [177]). The classic theory behind this is that a chromosome with one end broken during meiotic crossing-over can fuse with another such broken chromosome, leading to the formation of a “bridge“ conformation chromosome with two centromeres during the subsequent cycle of meiosis. This bridged chromosome is then ultimately cleaved into two daughter chromatids, but not necessarily at the site of the original breakage. This can lead to sequence deletion or replication on subsequently-healed daughter chromatids [176]. Experimental examination of this theory in *Caenorhabditis elegans* revealed evidence of such cycles, but also suggested more complex chromatin rearrangements can arise [178]. These more extensive rearrangements are proposed to arise from stalled replication events followed by template switching as may occur in areas with high-homology satellite sequences [178]. A simpler phenomenon is where non-reciprocal translocations can occur during break-induced DNA replication ([179], reviewed in [180]). Broken chromosomes are proposed to invade intact chromosomes with areas of homology during the G1 or G2 phase of the cell cycle, initiating DNA repair with the sequence from the other chromosome arm, possibly acquiring new genes and a telomere in the process [179]. Similar genome instability is also possible when telomeres are lost, making chromosome ends indistinguishable from double-strand breaks [181].

It is clear that telomerase and possible ITSs could have an important role in chromosome rearrangement. For example, when tobacco cells recovered to full cell viability after extensive chromatin fragmentation induced by cadmium stress, this was accompanied by a concomitant increase in telomerase activity [182]. Wheat chromosome end healing after gametocidal gene-induced breakage, efficient telomere healing by telomerase and stabilization of holocentric chromosomes in irradiated *Luzula elegans* plants were also previously reported [183,184]. Interestingly, when constructs containing telomeric arrays are introduced into mammalian or plant cells, the sites of integration become fragile, chromosomal breakage is induced and the new ends are stabilized [185,186,187]. Telomere-mediated chromosomal truncation has even been employed as a chromosome engineering technique [188,189,190,191]. All this supports the hypothesis that ITSs are preferred sites for breakage and that telomere-like repeats at a break site may favor chromosomal healing [170]. 

## 6. How to Find a Telomere Candidate

Methods and principles used for the identification of telomere minisatellites *de novo* substantially differ from those used for other satellites. Some details overlap, but there are several difficulties when comparing satellites and telomeres. For example, telomere repeat sequences do not possess recognition sites for restriction endonucleases and most telomere ends have single-stranded overhangs [192,193,194,195,196]. As a result, generation and cloning of blunt-ended fragments is often necessary. When genomic DNA is digested by restriction endonuclease, it is cut within subtelomeric regions resulting in so-called terminal restriction fragments (TRFs, Figure 2g,h). These consist of both telomeric and subtelomeric sequences up to the specific cleavage site for each chromosome end. Another more important complication is that, in most organisms, the number of repeats at any given end is not fixed, giving TRFs a typical heterogeneous or “fuzzy” appearance in Southern blots (see [75,128] for review and [197,198,199] for protocols, Figure 2g). No relationship between telomere length and genome size, chromosome size or chromosome number has yet been found. For example, up to 150 kb long telomere clusters occur in *Mus musculus* whereas telomeres of *Mus spretus* are only 5–15 kb long [200]. *Tetrahymena* cells contain two types of nuclei: the germ line micronucleus with five chromosomes and telomeres of >2.5 kb, and the transcriptionally active macronucleus with tens of thousands of minichromosomes terminated with 250- to 350-bp long telomeres [201]. Thus from a technical point of view, short telomeres like 50–300 bp in yeast and *Chlamydomonas*, andseveral kilobases in plants (e.g., *Silene*, *Arabidopsis*) and fungi, will not form a satellite band of distinct buoyant density that can be easily separated, unlike the α-satellites of guinea pig and kangaroo rat. This also makes it unclear which DNA piece to cut from a gel to use for cloning or to probe a library if telomere size is unknown. There is some tendency for unicellular organisms to have shorter telomeres as yeast, ciliate and green alga telomeres are mostly from 150 bp to several kb [30,128,142,202]. However, this simplification is not always correct, e.g., 300 bp, 20 kb and 25–80 kb long telomeres of *Chlamydomonas*, *Euglena* and dinoflagellate, respectively [30,202,203].

Today, various in silico approaches can be used to predict sequences prior to experimental verification (see next section, [146,204,205,206]). However, original experiments illustrate principles that are still valuable for unknown telomere identification and characterization. These can be employed when NGS data are not available or in silico analyses do not reveal a candidate. The following is a short summary of experimental approaches which have been used successfully in the past to characterize telomeres *de novo* (more examples and details about cloning strategies are summarized in [206]): 

(i) ***In vivo telomeres***. The story of the first identification of a telomere sequence [76] has been told many times (e.g., [207]), however, crucial for success was the choice of a ciliate model with thousands of telomeric ends in macronuclear DNA fragments. Linearized plasmids with *Tetrahymena* telomeric repeats at the ends in yeast [208] and later in fungi [209] confirmed the end-protection function of this newly-discovered sequence, something which enabled the construction of artificial chromosome vectors in yeast [210]. A similar strategy was employed for the isolation of telomeres from the fungus *Podospora anserina* [211] and transformation techniques with telomere capped vectors are now used in combination with state-of-the-art technology CRISPR/Cas for genetic manipulation (e.g., [212]).

(ii) ***C_o_t fractions***. Human chromosomes have much fewer chromosome ends than are found in the ciliate macronucleus. When telomere identification [49] is attempted in situations like this, a more manageable library containing likely telomeric sequences can be created from genomic DNA called the C_o_t fraction. Separation of C_o_t repetitive DNA fractions relies on the basic physical principles of denaturation and renaturation of dsDNA fragments [213] (Figure 1b). Briefly, sheared genomic DNA is denatured and single stranded fragments reassociated for a certain time at high temperature, so that sequences bind to complementary DNA. In principle, complementary strands of GC-rich and high-copy sequences will reassociate faster and in higher amounts than AT-rich and single-copy sequences. Longer times and lower temperatures allow reassociation of increasing proportions of dsDNA fragments. Excess unassociated single-stranded DNA strands are then removed, and the resulting blunt-ended dsDNA fraction can be used for library construction (Figure 1b). Moyzis and co-workers [49] identified bacterial colonies bearing plasmid constructs with telomeric sequences by hybridization with a human C_o_t fraction followed by re-probing with a hamster C_o_t fraction. Those which hybridized with both probes contained repetitive sequences shared by both genomes, in the case of human and hamster DNA this consists of Alu repeats, microsatellites and telomeric repeats. As C_o_t fractions represent mostly various high-copy sequences, unlabeled C_o_t DNA has also been used as a blocking agent to decrease non-specific background or to mark heterochromatin regions in FISH experiments [214,215,216,217]. Cloned C_o_t DNA fragments can also be used as a relatively cheap source of satellites and other repetitive sequence probes for karyotyping (e.g., [218,219], see [111] and references herein).

(iii) ***Bal31 nuclease sensitivity***. Bal31 nuclease progressively shortens DNA molecules from their ends. When applied to high-molecular weight genomic DNA, sequences that are at the original chromosome termini will be gradually shortened over time while internal genomic sequences, (e.g., rDNA, centromere, genes) will remain unaffected for some time (Figure 2h). After Bal31 digestion, DNA can be cleaved by restriction endonuclease and the resulting TRF separated by agarose gel electrophoresis. This principle was employed for determining the telomeric sequence of the model plant *A. thaliana. A. thaliana* has a relatively small genome (1C = 0.16 pg, [63]) of five chromosomes with relatively short telomeres [116], but otherwise few other repetitive sequences. A telomere enriched library was created (details summarized in [206]) and clones with known high-copy sequences (rDNA and 180 bp centromere satellite) were discarded. A final screening procedure was based on the expected sensitivity of telomeric sequences to exonuclease treatment. Laborious, but eventually successful Southern hybridization with pooled clones from the *Arabidopsis* library revealed one candidate (of 300 screened) which visualized typical shortening of TRFs and confirmed TTTAGGG as the telomeric sequence. From the very beginning to current telomere studies, TRF sensitivity to Bal31 nuclease treatment is a gold standard for the verification of the terminal position of a candidate telomere sequence (Figure 2h, on the right), especially in cases where cytological visualization at chromosome termini is not possible. A novel development of this experimental approach was where Bal31 sensitivity experiments were followed by generation of comparative NGS data to identify the unusual telomeres of *Cestrum* and *Allium* [146,205] (see below).

(iv) ***Crosshybridization with known telomere***. The sequence similarity among known telomeric repeats can be exploited in library screening experiments, allowing the use of a known telomeric sequence for the discovery of a similar, novel sequence. For example, a library for the identification of the *Chlamydomonas* telomere [202] was prepared as described in Richards and Ausubel [116] and screened for clones that hybridized with the *Arabidopsis* TTTAGGG probe. Further investigation revealed that these positive candidates contained the novel repeat TTTTAGGG rather than the *Arabidopsis* sequence and this sequence was confirmed as being in a terminal position by Bal31 digestion. The first insect telomere minisatellite, found in silkworms, was discovered in a similar fashion [115] when TRFs were visualized using a TTNGGG probe to set the initial detection conditions, followed by hybridization of clones with the same probe under low-stringency conditions allowing cross-hybridization. As a result of this, the sequence TTAGG was mapped at chromosomal ends and confirmed by Bal31 digestion. Crosshybridization has also been used as a selection principle for the identification of closely related telomeres in yeast and fungi [131,220], and telomere-enriched libraries have been generated by the end-cloning of TRFs (summarized in [206]).

(v) ***Telomerase products***. The last common experimental approach used to identify telomeres is the sequencing of telomerase synthesis products. Telomerase is a special reverse transcriptase that uses a part of its own telomerase RNA (TR) subunit as a template for the synthesis of telomeric repeats (Figure 3a,b, [207,221]). Telomerase binds at the 3′ end of telomeric DNA and synthesizes one repeat with each catalytic cycle. Originally, telomerase activity was demonstrated in a direct assay, using radioactively-labeled telomere oligonucleotides elongated by telomerase with the products resolved on polyacrylamide-urea sequencing gels [77,222,223]. The direct version of this assay works for organisms with high telomerase activity, but protein extracts with low activity cannot be detected this way and differences, e.g., among tissue samples, are often too small to reliably quantify. This problem was solved when the telomere repeat amplification protocol (TRAP, Figure 3c) was developed [224]. The TRAP assay employs the ability of telomerase to add telomeric repeats to the end of non-telomeric sequences, as occurs in chromosome healing and in ciliate macronucleus formation [225,226]. This addition is less efficient than for telomeric oligonucleotides but is sufficient to enable amplification of products by PCR. The non-radioactive variant of the TRAP assay (for protocols see [197,227,228]) produces fragments that can be cloned and sequenced rather than visualized with phosphorimaging. In principle, the synthesis of a candidate telomeric repeat by telomerase should unambiguously verify the true telomeric repeat sequence (Figure 3d,e). The obvious limit of this approach is the necessity to use a telomeric reverse primer with a candidate sequence in the PCR step, such that identification of a novel telomere repeat with a completely unrelated sequence to the reverse primer is not possible. However, this method was still successfully used to identify variants of telomere sequences in plants, algae, protists and insects [30,139,142,148,229,230], and can be used for both the verification of true telomeric sequences and to distinguish the correct variant between telomeric candidates (Figure 3d–f, discussion see below).

## 7. In Silico and Experimental Approaches Work Best in Combination

The development of NGS technologies resulted in a burst of genomic data however, repetitive sequences still represent a challenge for genome assembly algorithms (Figure 4). In particular, tandem repeats with low complexity and unit variability and the identification of sequence arrangement at chromosomal termini are hard to manage. For example, the telomeres of model alga *Chlamydomonas reinhardtii* were identified in 1990 [202], but the assembly of *Chlamydomonas* telomeric and subtelomeric regions was elucidated only recently [231]. Before the development of specific in silico tools and approaches, candidate telomere-like repeats across eukaryotes were identified manually (Figure 4c) in partially assembled contigs (series of overlapping sequences that form a continuous read) from available genomic datasets using simple searches for full or partial candidate sequences [30]. In some cases, genome assembly has identified candidate telomere sequences including the unusual telomeres of red alga [232]. Two telomere variants TTTAGGG and TT(T)CAGG were deduced from the genome assembly of *Genlisea* plants and were subsequently experimentally proven [233]. A current in silico approach suitable for routine use is Tandem Repeats Finder [234] which improves upon manual searches by identifying repetitive adjacent sequences within the same molecule and can use any assembled DNA contig, scaffold (sequences with gaps of a known size) or chromosome (Figure 4d). Similarly, RepeatExplorer is a computational pipeline which uses a bottom-up principle to investigate short NGS reads ([235,236], current update in [39]). RepeatExplorer is efficient for identification of various repetitive sequences like satellites, rDNA and mobile elements, but less so for short tandems of very low complexity like telomeres (see [206] for details, Figure 4e). Identification is much more difficult when telomeric sequences do not match known or expected telomeric types. More details on these and other in silico methods are provided below (Section 9.4).

In the past, the attempts to validate candidate repeat sequences considered structural features that seemed to unify telomeres, namely the guanine quadruplex (G4) and telomeric loop (t-loop). The first telomere sequences identified in ciliates (TTGGGG and TTTTGGGG) form G4 structures in vitro and much work has gone into investigating G4 formation in other telomere variants [132,237,238,239], summarized in [10]). Typically the terminal part of the G-rich strand of the telomere is a single-stranded overhang in vivo [192,193], experiments to detect G4-structures in telomeres have had variable results to date (see [10] for review). More success has been achieved in the search for G4-binding proteins although the biologically relevant G4-structures investigated are not exclusively telomeric (see [240] for review, [58,241,242]). The power of in silico tools for the prediction of G4 formation [243,244] is limited because the universality of this structural feature in telomeres has been questioned experimentally [239] especially in the case of telomere repeats with cytidine residues from nematodes (TTAGGC, [245]), arthropods (TTAGG/TCAGG, [115,143,246]) and a nucleomorph of chlorarachniophytes (TCTAGGG [247]). Telomere protection via formation of a t-loop seems to be more conserved across a range of telomeric sequences, having been demonstrated in *Caenorhabditis elegans,* garden pea, animals and yeast ([196,248,249,250], see [9] for review).
Figure 4**NGS approaches in the search for the telomere motif candidates.** (**a**) Classical experimental approaches like FISH, TRAP, and TRF sensitivity to BAL31 nuclease are equally useful for initial screening or for verification of in silico results. Similarly, when telomere DNA signals are below detection limits or a species possesses an unusual telomere sequence, NGS data can be used to search for telomeric motifs. (**b**) NGS is a good first-choice method due to genomic data availability and low price. (**c**) Reads assembled from raw data to long contigs and scaffolds are easily inspected manually for classical telomeric types with a C-rich 5′end and a G-rich 3′end. (**d**,**e**) Unassembled data can be searched for likely telomere candidates, either abundant minisatellite repeats (**d**) or various repetitive sequences (**e**). Terminal position of candidate sequence should then be verified experimentally as in (**a**). (**d**) Representative outputs from Tandem Repeats Finder ([234], https://tandem.bu.edu/trf/trf.html, accessed on 22 August 2022). Telomere candidates were identified as predominant minisatellite repeats in yeast *Lachancea* sp. [132], beetle *Anoplotrupes stercorosus* [151], a plant with human-like telomere sequence *Zostera marina* (Alismatales) [251] and also a plant with unusual telomere type *A. cepa* [146]. The predominant minisatellite TTTTTAGCAGT in *Cestrum* is a high-copy sequence associated with various repeats (e.g., BR23, Figure 2d, [157]) and masks the real telomere repeat TTTTTTAGGG [205,206]. (**e**) The pipelines of RepeatExplorer and TAREAN ([39], https://repeatexplorer-elixir.cerit-sc.cz/, accessed on 22 August 2022) are designed for long repeats. This can permit identification and characterization of longer or unusual telomere candidates from NGS data, such as the alternative telomeres of *Anopheles*, *Chironomus* and *Drosophila* [11]. The graph-based analysis detects long satellites and LTR-retrotransposons as rings and circles. (**f**) The vast majority of eukaryotic telomere repeats are minisatellites with consensus sequence T_x_A_y_G_z_ like the insect-type core motif TTAGG (bold red). Longer minisatellites tend to have more sites with such conserved stretches (see highlighted nucleotides). (**g**) Novel bioinformatic approaches use prediction of TR subunits in combination with results from Tandem Repeats Finder to identify telomeric repeats synthesized by telomerase as described in detail in [132,204,252] (animal and yeast TR were identified previously, see [134,253,254] and references herein).
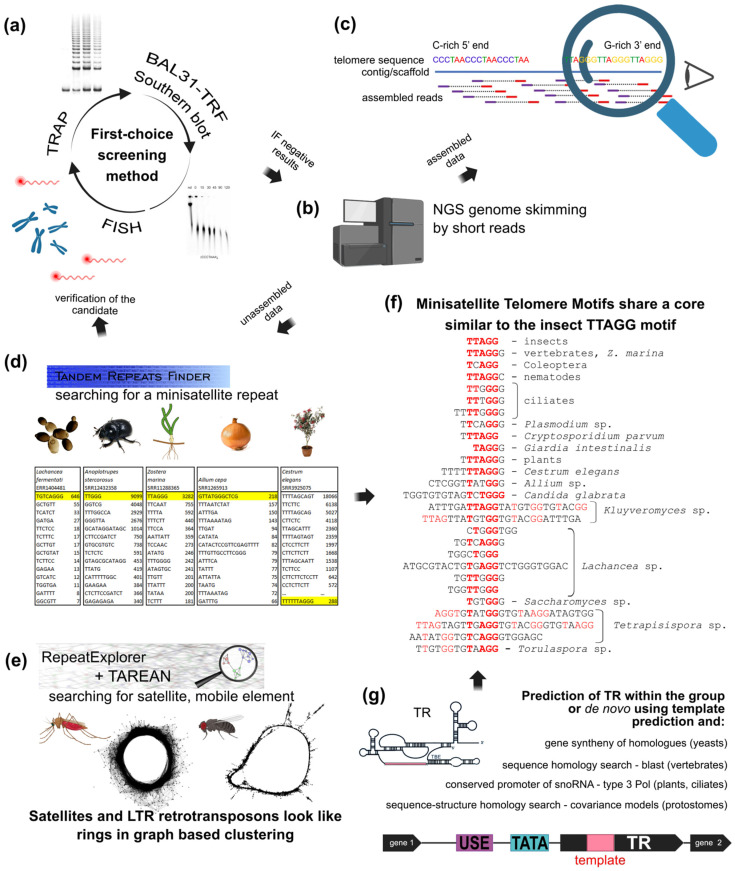



T-loops are lariat structures in which the 3′ overhang invades an upstream double-stranded section of the telomere, forming a loop of several kilobases of DNA. In vivo trapping and visualization of t-loops also identified related extrachromosomal DNA circles, termed t-circles (see [255,256] for review and references herein). T-circles were originally found in the linear mitochondrial chromosomes of the non-model yeast *Candida parapsilosis*, but much work followed soon after when similar structures were visualized in human telomerase-negative ALT cells [257,258,259,260]. Further work with these cells identified that t-circles were dsDNA resulting from cleavage of t-loops, prevalent in ALT cells which use an alternative form of homologous recombination–driven telomere elongation [261,262]. Later work also identified small G-rich or C-rich ssDNA circles [263]. T-circles can provide substrates for rolling circle–dependent amplification and in turn recombinational telomere elongation by various tandem repeats [262]. Thus, t-circles may represent a relic of an alternative pathway for telomere maintenance [9,256]. T-loop prediction from sequence data is not currently possible in silico, and although prediction of G4 structure formation as a parameter for candidate telomere sequences has been investigated [132], such studies currently await experimental validation.

One more feature can be used for validation of in silico predictions of telomere repeats and that is the telomerase RNA subunit. Primary sequences of TRs are known to diverge significantly, even between closely related species. In general, the only features that seem to unify TRs are an evolutionarily conserved secondary structure called the pseudoknot and a small segment which dictates the telomere repeat synthesized by telomerase called the template region (reviewed in [264]). The pseudoknot structure is difficult to predict in silico and so usually it is modeled after the identification of the TR sequence. Similarly, finding the TR template region in NGS data is a challenge because known TR templates are relatively short, with sequences slightly longer that the length of repeat unit (Figure 3a,b). Some of the greatest challenges when identifying novel telomere repeats and/or TRs *de novo* can occur when a sequence is completely unrelated after an evolutionary switch, as in *Allium*; divergent telomeric sequences as found in yeast; or alternative mechanisms of telomere maintenance as in *Drosophila* and *Chironomus*. However, with an open mind research into non-model systems is certainly possible. An example of this is the identification of the *Allium* telomere [146] followed by the *Allium* TR and in turn the identification of plant TRs and telomere sequences [204,252] using a novel strategy that combined experimental genomic, transcriptomic and in silico approaches (see below).

Today, raw data or assembled contigs in public NGS datasets exist that can routinely be accessed and mined for repetitive sequences using in silico tools. In some cases, specific experimental designs have been developed that could serve as an inspiration for future research. The following is a short summary of successful examples and new ways of in silico analysis which have been used in combination with experimental approaches for the identification and verification of novel telomere sequences.

***(i) Identification of genomic TR locus in yeast.*** One of the factors that makes searching small genomes for TR loci difficult is the low similarity in primary sequence among TRs. A known TR (*TLC1* from budding yeast) synthesizes irregular repeats TG_1–3_, and a direct amplification of the sequences from the related species proved unsuccessful. However, protein encoding sequences bordering the *TLC1* loci are more conserved and were instead used for primer design so that cloning of the complete *TLC1* loci was possible in close relatives of *S. cerevisiae* [265]. A simple BLAST search using the sequence of 1.5 telomeric repeat (5′-GTTAGTCAGGGTTAG-3′) as a query was used to successfully identify the genomic TR loci in *Yarrowia lipolytica*, and subsequently in *Yarrowia* clade species, defining these divergent telomeric sequences from their template regions [133]. This complex study by Cervenak et al. [133] represents an example of good practice combining in silico and experimental approaches leading to a detailed description of the co-evolution of TRs, telomeric repeats and telomere-binding proteins in yeasts. A similar search protocol was used for TR loci identification from relatives of filamentous fungi *Aspergillus oryzae* [266]. Such TR sequence data can also be used in combination with other techniques to confirm telomeric sequences, for example a recent focused identification of TR subunits was used as an additional parameter to support the identification of the highly variable telomere repeats in Saccharomycotina [132].

***(ii) Combination of NGS and Bal31 digestion.*** From 1995 and 2003 *Cestrum* and *Allium* respectively were known to have unusual telomeres [114,135] but experimental approaches struggled to identify true telomeric repeats [135,139,140,156]. *Allium* and *Cestrum* genomes are huge (1C = 17.9 pg for *A. cepa* [106] and 1C = 10.9 pg for *C. parqui* [135]) with the prevailing chromosome number n = 8 in haploid nuclei. Thus, each chromosome has two ends, but nearly the same number of nucleotides as an entire animal genome (e.g., 1C = 3 pg, n = 23 in human, [267], see http://www.genomesize.com, accessed on 1 August 2022, [268]) or relative plant species (e.g., *N. sylvestris*, 1C = 2.7 pg, n = 12, see https://cvalues.science.kew.org/, accessed on 1 August 2022, [63]). In silico tools did eventually identify the enigmatic telomeres in *Allium* and *Cestrum* plants [146,205], but this required considerable experimental effort before NGS data were generated. As a result, a general strategy to find telomere sequences in eukaryotes, with particular utility to species with large genomes, was developed (see [206] for details). Briefly, high-molecular-weight DNA was split into two pools, one of which was treated with BAL31, and NGS data of these two DNA pools were compared to find any repeat that was under-represented in the BAL31 digested dataset, which would likely be a candidate telomere. The bioinformatic tool RepeatExplorer [236] successfully found a candidate sequence for *Cestrum* data [146] but not for *Allium*. To strengthen the approach, BAL31-NGS datasets from 11 *Allium* species were compared to find overlapping repeats identified using Tandem Repeats Finder [234] (Figure 4d). Tandem Repeats Finder is suitable for very low complexity sequences and was trained for the analysis of these data (details in [206]). This approach also identified a candidate and finally the terminal positions of candidate sequences, TTTTTTAGGG in *Cestrum* and CTCGGTTATGGG in *Allium*, were confirmed experimentally [146,205].

***(iii) From a comparative transcriptome study to a new bioinformatic approach.*** Building on the work in (ii), the template region of TR can be used to predict telomere repeats synthesized by telomerase and vice versa (Figure 4g). An example of this concept is a broad study to identify TR and telomeric repeats across the entire land plant phylogeny [204]. This story starts with the identification of onion TR. Conventional datamining of *Allium* genomic sequences in search of the TR subunit is prohibitive because of the *Allium* genome size. Identification of TR was instead achieved by comparison of transcriptomes of distantly related *Allium* species. The strategy for this exploited the telomere repeat shared among *Allium* species which is both unusual and long [146]. TR subunits, which serve as a template for telomere synthesis, must contain this sequence and so were identified using this as a search parameter. Then using a combination of newly generated and publicly available transcriptomic datasets, more TRs were identified from Asparagales plants synthesizing human-type telomeric repeats (e.g., *Tulbaghia*, *Asparagus*, *Agave*) and from the Asparagales that have *Arabidopsis*-type telomeres (the orchids *Dendrobium* and *Phalaneopsis*). These results in Asparagales encouraged a further search in public datasets from representative species of major clades across land plant phylogeny that resulted in the identification of 75 TR subunits, including common model plants (e.g., *Arabidopsis*, *Nicotiana*), crop plants (e.g., garden pea, maize, rice, olive, sunflower, grape, kiwifruit, coffee, cocoa) and plants with unusual telomeres (*Cestrum*, *Genlisea*). Moreover, this wide comparison of TR candidates found within genomic datasets that delve all the way to the root of land plant phylogeny, solidified the characterization of conserved TR gene regions with putative regulatory functions, i.e., TATA box, Upstream Sequence Element (USE) and terminator. The plant TR gene was found to be highly conserved in contrast to the far more divergent TR genes found in animal and yeast. Moreover, this enabled a relatively easy bioinformatic way to identify TRs and to predict telomere DNA sequence in virtually any land plant species with unknown telomeres. Experimental validation of TR function from several plants proved the reliability of this method of candidate TR identification. As such, there is now a relatively easy bioinformatic way to identify TRs and to predict telomere DNA sequence in virtually any land plant species [204] (Figure 4g). Interestingly, the promoter characteristics (TATA and USE) classified plant TR as a product of a Pol III-dependent pathway similar to ciliate TR synthesis suggesting an even deeper common origin of TRs [204]. In principle, this opened up the possibility of identifying novel telomere repeats in plants or other organisms, not from DNA only, but through identification of USE and template regions of TRs. Indeed, recently this new bioinformatic approach led to a broad identification of telomere sequences in green algae, ciliates and Stramenopiles including novel types TATAGGG, TGTTAGGG, TGTAAGGG and demonstrated the deep evolutionary TR origin in the megagroup Diaphoretickes [252]. 

## 8. Telomere Proteins

Chromosomal DNA in cells associates with proteins that fold these long polymeric molecules into condensed, ordered forms. Most of the DNA sequence, including genes, subtelomeric satellites and the proximal sections of telomeres is folded into a series of compact but dynamic protein-DNA complexes called nucleosomes [269]. In 2001, Fajkus and Trifonov [270] proposed telomeric nucleosomes are packed in a variant, columnar chromatin structure. Recently, the formation of this structure was confirmed experimentally using cryoelectron microscopy [271]. The ends of telomeres associate with a more diverse set of proteins depending on organism that maintain a 3′ single-stranded overhang (aka G-overhang), recruiting enzymes to lengthen the 3′ strand and shorten the 5′ strand which induce and stabilize t-loop formation (reviewed in [9,272]). These mechanisms protect telomeric DNA, prevent aberrant DNA repair and mediate interactions with telomerase (see above, [272]). In *Arabidopsis* and *Chlamydomonas* some telomere ends are instead blunt, with no or little 3′ overhang, although it is unknown whether this is a special feature of these organisms or a more widespread characteristic [195,273,274].

Of principal interest to telomere researchers are the specialist proteins that interact with the distal sections of telomeres at the ends of chromosomes. Two major telomere protecting complexes have been described, CST and shelterin. These were initially thought to be alternative mutually exclusive systems, but the search for homologues revealed that many eukaryotes, including humans, had both systems able to work in parallel [275,276,277,278]. Continuing research focused on looking for homologues of human systems across all eukaryotes, however this approach has had only partial success (reviewed in [279]). The CST complex is largely conserved in eukaryotes [280] in terms of function, if not necessarily the sequence of its components [281,282,283]. CST binds ssDNA and recruits Pol1α primase for C-rich strand synthesis and also has a role in preventing stalled replication forks (for recent advancements see [278] and references herein). In comparison, shelterin (reviewed in [272]) coats telomeric DNA generally and interacts with telomerase for G-rich strand synthesis. Shelterin is not present in all eukaryotes, although most have an identifiable protein family that occupies the same role (Figure 5). In addition to these larger end-protection protein complexes, there is a highly conserved heterodimer of proteins called Ku70/Ku80 that is normally involved in non-homologous end-joining events, but which also has an enigmatic role in telomeres. This complex binds dsDNA ends non-specifically, but is known to interact with components of shelterin in mammals and telomerase RNA in yeast (reviewed in [284,285]). 

Shelterin has a dynamic composition and variant complexes bind the entire length of distal telomeres, there are six core protein components in humans which are more-or-less thought to be conserved in mammals [272,288]. Telomeric repeat binding factor 2 (TRF2) binds the telomeric DNA motif with nanomolar affinity via a SANT/Myb domain sometimes termed the telobox in older literature [289], not to be confused with interstitial telomeric motifs, which are also called teloboxes [172]. TRF2 binds dsDNA and homodimerization enhances this process. It is also proposed to have helicase-like activity where it can wrap dsDNA from near to the telomere end around itself causing steric torsion in the telomere end that encourages T-loop formation. Consistent with this, TRF2 is both necessary for T-loop formation by shelterin and capable of forming T-loops in the absence of any other shelterin components [290]. TRF1 is a highly homologous protein to TRF2 which only binds telomeric repeats and lacks T-loop forming ability. Both proteins (possibly as homodimers) bind TRF1-interacting protein (TIN2) to form the core dsDNA binding subunit of shelterin [291]. TIN2 binds TINT1/PIP1/PTOP1 (TPP1) which in turn binds protection of telomeres 1 (POT1), a protein with multiple OB-fold domains that can bind ssDNA and which is thought to be the main interactor with the 3′ overhang in the complete shelterin complex. TRF2 alone can also recruit repressor/activator protein 1 (RAP1) as the sixth member of core shelterin and the interactions between shelterin subunits can generally occur across multiple protein surfaces [291]. TPP1 in complex with POT1 interacts with telomerase as part of the coordination of telomerase and shelterin protein complexes [278,292].

Unsurprisingly, *Drosophila* has evolved a separate group of proteins in a complex called terminin to protect the retrotransposon-derived sequences at the ends of its chromosomes. Terminin was identified from the larval brain cells of mutant flies with end-fused chromosomes and consists of a core of heterochromatin protein 1/origin recognition complex-associated protein (HOAP), Modigliani (Moi) and an OB-fold protein called Verrocchio (Ver) [293]. Whilst fission yeast has a shelterin complex made from paralogues of human proteins [294,295], budding yeast, instead has a velcro-like network of proteins called the telosome. This consists of Rap1, a general transcription factor which coats double-stranded telomeric DNA, Rif1 which binds DNA via a Myb domain and Rif2 which binds DNA via an AAA+ domain. Rif 1 and Rif2 can bind four or two molecules of Rap1 respectively through binding domains attached to long disordered chains to form a dense protein network ([296], reviewed in [297]). The system in plants is not yet clear (reviewed in [298]). 

Although plant proteins that share some sequence homology to human shelterin proteins have been identified (summarized and reviewed in [299,300]), including those with C-terminal Myb domains similar to TRF1 and TRF2, these do not have any obvious end-protection role [301]. The only definitive double-stranded telomeric DNA binding proteins so far characterized in plants are the telomere repeat binding proteins (TRB1–3) [302,303,304]. These proteins bind to *Arabidopsis* telomeres in vivo [304,305], and TRB1 colocalizes with telomeres when introduced to *Nicotiana benthamiana* in live cell imaging studies, suggesting a general role for these proteins at plant telomeres [306,307]. TRBs have histone-like domains that allow multimerization and binding to telobox-related DNA motifs in a multitude of chromosome sites and N-terminal Myb domains that specifically bind double stranded telomeric DNA [302,303,304,308,309]. Similar to TPP1 in human shelterin, TRBs can interact with telomerase and so together with DNA binding and multimerization it is easy to draw parallels with other end-protecting proteins [289,299,303,304]. It can be speculated that in addition to their other regulatory DNA-binding roles [308,309,310], TRBs could form some sort of end-protection framework, similar to the telosome in yeast. Alternatively, it could simply be that any end-protection proteins in plants are sufficiently variant from other organisms to have eluded discovery so far. 

One final quirk in plant telomere biology is the occurrence of blunt-ended telomeres. Some blunt DNA ends in *Arabidopsis* [311] are known to at least temporarily bind Ku70/80, a ubiquitous DNA end-protecting protein complex that is part of the normal double-strand break maintenance mechanism. Studies in budding yeast and human cells revealed that Ku can interact with telomeric chromatin either by directly binding to telomeric DNA or via interaction with telomere associated proteins, including the shelterin subunits such as TRF1, TRF2 and Rap1 [312,313,314]. Studies using mice revealed considerable telomere abnormalities where Ku is knocked out, but phenotypes are complex enough that a specific role is difficult to ascertain [315,316]. In yeast, Ku also binds the telomerase RNA *TLC1* separately from telomere ends in a mutually exclusive fashion, and is required to maintain levels and nuclear localization of *TLC1*. YKu association with telomeres is independent of its association with *TLC1* RNA and occurs throughout the cell cycle [317,318]. As with other eukaryotic systems, the Ku heterodimer in *Arabidopsis* forms a tube that slides onto and encircles the double-stranded telomere from one free end, providing simple end-protection without translocating inward [274,311]. It is so far unknown whether Ku-protected blunt ends in *Arabidopsis* and *Chlamydomonas* are unique to these organisms or whether a more widespread phenomenon is yet to be found in other eukaryotes. It is possible that these are an evolutionary step that limits the amount of work that telomerase has to conduct or provides cells without telomerase more stability during proliferation [319].

## 9. Telomeres in Practice

Seeing is believing. In this section we briefly describe selected experimental approaches with focus on those that can be used for the identification and verification of candidate telomere repeats (further telomere biology methods are discussed in [197]). Using examples, we especially focus on possible problems and their solutions when proving the terminal position of a candidate telomere. We also briefly document several pros and cons of probes and techniques that can affect the interpretation of results.

### 9.1. In Situ Hybridization

FISH is often the first choice of method used to investigate telomeres (or other micro/minisatellites). Conventional fluorescence in situ hybridization (FISH) uses oligonucleotides, cloned fragments or concatemers as probes, similar to Southern hybridization. Probes for FISH can be labeled directly with a fluorophore or indirectly using biotin- or digoxigenin-modified nucleotides allowing for secondary detection (Figure 6a). Sequence specificity and detection limits can be improved using peptide nucleic acid (PNA), locked nucleic acid (LNA) probes or the primed in situ (PRINS) labeling reaction ([123,320,321], reviewed in [120,197]). In principle, the advantage of PNA and LNA probes over conventional DNA oligonucleotides or dsDNA is their strong binding to a target nucleic acid, allowing for more stringent washing conditions which lower the background and improve the detection limit. The disadvantage of these is their cost and as such conventional probes are frequently used. Generally for in situ hybridization, if the ends of chromosomes are clearly marked, this can be taken as the first proof of telomere sequence identity (Figure 6d). When results are negative, cloning and/or in silico prediction can be used to search for candidate telomeric variants that can in turn be tested. The example of telomere identification in dinoflagellates (example (i) below) illustrates the power of this a first-choice experimental approach for suitable species. Example (ii) uses principally crosshybridization and demonstrates how some insight into telomere structure is achieved using genome in situ hybridization (GISH). Examples (iii) and (iv) show potentially misleading FISH results that may result in misidentification of telomere sequences due to the detection limits of the method.

**(i) *Test by chance.*** Numerous studies of plant, algal and insect telomeres have shown that variations in telomeric sequences tend to be relatively modest, and often feature the same sequences. For example, the human-type (TTAGGG) and *Arabidopsis*-type (TTTAGGG) telomeric repeats emerged independently across the tree of life [30] and they group phylogenetically related species [142,148]. As such, when exploring new species it can be worthwhile to test common telomeric sequences from other organisms as a first step in hybridization studies. One example is a karyotyping study by FISH of the algal group Dinoflagellates using a probe containing the *Arabidopsis* consensus telomeric sequence that marked chromosomal ends of *Prorocentrum micans* and *Amphidinium carterae* [323]. Dinoflagellate genomes are the largest in the Eukarya domain, these chromosomes lack histones and may exist in liquid crystalline state. As such, it was important to follow up FISH studies with other methods (described below) to confirm telomere synthesis by telomerase and terminal position of TTTAGGG repeats, as was proven in *Karenia papilionacea* and *Crypthecodinium cohnii* [203].

**(ii) *GISH shows overlap.*** The GISH technique was originally employed to distinguish parental genomes in plant hybrids ([110,324], see [325] for review, an example of GISH technique in allotetraploid *Cardamine scutata* is shown on Figure 2, [159]) and takes advantage of the fact that genomes are full of repetitive sequences. The whole genome of one species is used as a probe which marks repeats which are shared in close relatives, but are not present in evolutionary distant members of the same genus [110]. As increasingly distant relatives are hybridized, only two types of shared sequences remain detectable, rDNA and telomeres [326]. GISH can be a good approach to hint towards novel telomeric repeats where conventional FISH with known sequences has failed. This approach was used in *Allium* where an *A. cepa* probe showed strong signals for the positions of both rDNA repeats and all chromosome ends of the distantly related *A. ursinum*. GISH detection in this case did not identify the telomeric sequence but showed that chromosome termini of distant relatives share conserved sequences [146].

**(iii) *Multiple telomere variants are present and their mutual position at termini cannot be distinguished by FISH.*** Adams and co-workers [136] found representatives from 12 plant families that were not terminally labeled by the typical plant telomeric sequence clustered in a derived clade within Asparagales. They predicted that there was a single evolutionary event when the *Arabidopsis*-type telomere sequence was lost in early progenitor of these families that comprises up to 6300 species (ca. 2.5% of angiosperms). However, four species in *Ornithogalum* (Hyacinthaceae, Asparagales) showed signals of typical TTTAGGG sequence at some, but not all ends [136]. The family Hyacinthaceae has a central position within Asparagales. Further research and cloning of TRAP products revealed that the human-type telomeric sequence replaced the typical *Arabidopsis*-type telomere repeat in these families including Hyacinthaceae [135,327,328] (Figure 6). In most cases the human-type variant occurs along with a lower abundance of the *Arabidopsis*-type and *Tetrahymena*-type variants of the minisatellite sequences and FISH showed that the variants can occur mixed together at the telomere ([139], Figure 6e,f). More telomere sequence variants were also observed in human telomeres [329].

**(iv) *Cloned probe was not telomeres***. Within Asparagales, species of the genus *Allium* lack both the *Arabidopsis*-type telomeric sequence [114,136] and human-type sequence [139,140]. These species have huge chromosomes and attempts to identify the telomere sequence using microdissection and end-cloning approaches [161] revealed as candidates the ACSAT satellite, rDNA and transposable elements. FISH targeting ACSAT seemingly marked all chromosomal ends of *Allium* species in section Cepa, close enough to the end that the sub-telomeric position was not clear [51,105]. However not all *Allium* species possess this repeat (Figure 1j) and this question remained open until 2016 when Fajkus, Peska and co-workers identified and proved the unusual *Allium* telomeric sequence (CTCGGTTATGGG) using a BAL31-NGS approach (see above) [146]. Thus, the high-frequency repeats overwhelmed the true telomeric sequence in FISH experiments.

### 9.2. Southern Hybridization, gDNA Isolation and Bal31 Nuclease

Many species are not accessible to in situ hybridization techniques and detection at chromosomal level, but other hybridization methods can still be used to distinguish telomeres, subtelomeres and other satellites at molecular level. The power of Southern hybridization lies in the possibility to investigate a large number of samples [138,142,148,151,330,331]. This technique gives information about the presence or absence of a tested sequence in samples, but has to be combined with other techniques to prove the terminal position of telomeric candidate sequences and their function, e.g., Bal31 digestion combined with TRF method (see [197,198,199], Figure 2h above) and/or TRAP product sequencing (Figure 3). After restriction digestion, Southern hybridization can distinguish TRFs and additional signals originating from ITS sequences (Figure 2g). TRF signals can be visualized, measured, compared and statistically evaluated by user-friendly webtools, e.g., WALTER (see [332] and references herein). For species with small genomes (or short telomeres), collections of restriction endonucleases should be tried to get a full picture of TRF signals that could represent individual chromosome ends suitable for further mapping (e.g., [202,333]). As with in situ hybridization, oligonucleotides, PCR-amplified concatemers or plasmids with cloned telomere sequences can be used as probes for telomere sequence variants. The merits of different probes and some recommendations are detailed in [322]. Briefly, oligonucleotides are easy to access, relatively low cost, and their precise, user-defined sequence is ideal for screening. PCR-amplified concatemers were introduced by Idjo et al. [334]. Concatemers are generated by self-annealing of C- and G-telomeric oligonucleotides and amplification by Taq DNA polymerase in template-free PCR reactions (Figure 6b). Obviously, PCR products represent a mixture of telomeric dsDNA fragments of different length which can be optimized using certain PCR conditions [198,322]. Despite concerns of potential amplification mistakes and cross-hybridization, false positive signals are more prevalent for oligonucleotides than concatenated probes [322]. Plasmids with cloned telomeric fragments are often used as probes, cloning of concatemers is also possible. However, such plasmids should be checked occasionally for the presence of the correct insert as we found plasmids with cloned concatemers that were prepared in our laboratory in 2003 were not stable when propagated in *Escherichia coli* (unpublished result Sykorova). As other plasmids with cloned natural telomeric inserts have been used successfully by other researchers, e.g., the pAtT4 from *Arabidopsis thaliana* that contains 53 perfect repeats (TTTAGGG) and two variant repeats (TTTAGAG) [116], we speculate that some sequences or vectors are better accepted by bacteria than others.

When verifying the position of a candidate sequence on chromosome termini at a molecular level, it is noteworthy to consider the quality of gDNA preparation. Bal31 digestion followed by TRF analysis is usually performed using high-molecular-weight DNA samples (see in [206]) and pulse field electrophoresis, but shorter fragments can be visualized using conventional agarose electrophoresis. High-molecular-weight DNA samples are preferred for Bal31 digestion because the enzyme also shows endonuclease activity at dsDNA sites with altered structures and exonucleolytic degradation of ssDNA (and the same for RNA, Figure 2h). These activities result in more DNA ends to be shortened thus, the quality of starting DNA samples is critical. Samples treated by Bal31 nuclease were used previously for dot-blot hybridization, although some experimental factors should be considered for this to be effective. Firstly, DNA samples should be of good quality with molecules as long as possible and without apparent degradation. Preparation of genomic DNA samples from non-model organisms often requires optimization [30] to avoid natural contaminating compounds that are not visible on gels, but interfere with digestion. As such, it is worthwhile to check the digestion of gDNA samples before experimentation, e.g., by detection of TRF fragments and/or using a universal probe (rDNA) as an internal control and monitoring gDNA integrity across multiple time intervals. Other factors worth considering when telomere length and sequence is not known: Telomeres should be remarkably shorter than DNA molecules in samples to avoid false results for sensitivity to Bal31 digestion, there should be homogeneity in repeat sequences to avoid false results caused by crosshybridization with telomeric variants and if clusters of ITS sequences are present, these can increase the background. Genomic DNA prepared by standard isolation methods, including commercial kits, usually produces a mixture of randomly sheared DNA fragments of ca. 20–50 kb in length. Sequence-specific qPCR amplification of control genomic sequences revealed that such gDNA samples showed an initial ca. 40% drop in amount of genomic DNA after short-term Bal31 digestion [162] (Figure 2i). After this, the decreasing DNA level of all internally located genomic sequences equally reflected the progression of the BAL31 digestion. In addition, processive shortening of gDNA was monitored using unique TAS sequences that diminished in relation to their distance to telomere. This principle was demonstrated using subtelomere PCR/qPCR in *Arabidopsis* [162] (Figure 2j,k) and can be used for the verification of specific telomere-subtelomere border regions or of unique TAS sequences in genomes with short telomeres.

### 9.3. TRAP Products

Crude or telomerase enriched protein extracts from potentially telomerase active tissues can be tested using the TRAP assay [197,227,228]. Crude extracts are often the first choice and when little or no telomerase activity is detected, telomerase enriched extracts can be prepared by selective PEG precipitation. Protein precipitation by PEG is not specific for telomerase, but enrichment of telomerase was observed in precipitated fractions of plant extracts [197,335] and has been successfully used in many other species. In our experience with diverse sample collections, some algal species do not show enrichment in PEG precipitated fractions in comparison to crude extracts, thus it is worth initially retaining the soluble fraction for testing in parallel [30,142]. The TRAP assay is very sensitive to contamination by adventitious eukaryotes possessing their own telomerases (usually fungi, mites or protozoa) which can cause false positives. In the case of algal strains from culture collections [30,142,148], all samples were carefully examined and only axenic strains or those with bacterial contamination were investigated.

One of the strengths of identification of telomere variants by TRAP assay derives from the fact that a reverse primer with a limited sequence similarity can still be effective for crosshybridization with telomerase-elongated products (examples in Figure 3d–f). This was demonstrated in Asparagales [139], where a reverse primer with the 7-nt Arabidopsis-type telomeric repeat was used, and a pattern with shorter periodicity was produced (Figure 3d). This clearly indicates the synthesis of a variant repeat which could be verified from cloned TRAP products. In the case of Asparagales, the human-type repeat was found. Interestingly, investigation of *Daphnia* detected a variant repeat type when a commercial TRAPeze kit for investigation of human telomerase activity was used [229]. A ladder of TRAP products with shorter periodicity than in the control was observed and the prevalent insect-type of repeat TTAGG was identified in TRAP products (see Figure 3f). Another example is from the streptophyte alga *Klebsormidium* which produced a ladder with the same 7-bp periodicity as the typical *Arabidopsis*-type TTTAGGG, but the variant sequence TTTTAGG was revealed after cloning and sequencing of TRAP products [30]. Absence of TRAP products can be caused by (i) low telomerase activity in the protein extract tested, (ii) no similarity to the sequence of the reverse primer, (iii) limited binding of telomerase to the substrate oligonucleotide. For (ii) and (iii), we recommend testing of reverse primers with sequences corresponding to alternative telomere types or minisatellite variants and/or using different substrate primers to cope with possible telomerase substrate preference.

### 9.4. In Silico and Experimental Approaches

Many researchers find it expedient to start with in silico prediction of satellite sequences (Figure 4b–e) rather than the first-choice screening methods (FISH, TRAP, BAL31 TRF, Southern blot; Figure 4a). Typically, experimental approaches can then be used to verify these predictions. Equally, in silico methods can be a valuable way of confirming experimental findings. As well as classic Next Generation Sequencing (NGS), other sequencing platforms are available. Third Generation Sequencing (TGS) platforms led to a breakthrough in sequencing extremely challenging repeats such as long blocks of microsatellites. Up to several hundreds of kbp (hypothetically even several Mbp) can be sequenced per TGS read and all kinds of satellites can then be visualized directly, e.g., in dot-plots [336]. Oxford Nanopore Technology (ONT) platforms raw reads with 98.3–99.92% accuracy, depending on the library kit, flowcell type and data postprocessing used (https://nanoporetech.com/accuracy, accessed on 1 August 2022). Accuracy can be further improved by preparing consensus sequences with sufficient target coverage. Pipelines for tandem repeat mining from ONT data are detailed further in [40]. Due to these developments, numerous genomic and transcriptomic datasets have become available and can be used to conduct searches without the need of additional sequencing. For example, public databases administrated by the NCBI are a reliable source of primary data (SRA from NCBI, https://www.ncbi.nlm.nih.gov/sra, accessed on 1 August 2022). Manual evaluation of fully or partially assembled sequences such as contigs is possible using the genome collection (NCBI Genome https://www.ncbi.nlm.nih.gov/genome/, accessed on 1 August 2022). When doing so, it is noteworthy to mention that telomere sequences are strand oriented, i.e., the G-rich strand sequence should be at the 3′ end and the C-rich strand sequence at the 5′ end of the DNA molecule (Figure 4c). When G-rich telomeric minisatellites occur internally or at the 5′end of a contig, this is either an ITS or a misassembled piece. It is not necessary to have assembled data for telomere sequence identification. In order to datamine computed unassembled raw reads, Tandem Repeats Finder, RepeatExplorer and its TAndem REpeats ANalysis (TAREAN) tool (Figure 4d,e) are recommended. The preferred raw data are Illumina paired end reads without overlaps (e.g., read length 150 bp, library insert size > 300 bp). The Illumina platform HiSeq X Ten was reported to underrepresent low complexity sequences, including telomeric sequences, and so is not recommended [337]. To quantify the genome proportion of repeats it is recommended to use PCR-free library preparations and at least three technical replicas. A good place to start is to analyze datasets from a species of interest together with close relatives. As species tend to share repeats including telomeres, it is advantageous to look for both overlaps between species and exceptions from the rule. Notably, from a broad phylogenic perspective, telomere motifs and telomerase products differ greatly in sequence and length. Often the diversified motif has a conserved core similar to the shortest telomere unit found in insects, TTAGG (Figure 4e). 

Satellites (including telomere minisatellites) and other repetitive sequences can be used to build phylogeny and provide a different perspective on taxonomy [112]. RepeatExplorer [39] is a computational pipeline that uses graph-based clustering of next-generation short sequencing reads. Clusters are further used for identification of consensus repeat sequences and comparison between samples or species is possible. In contrast to Tandem Repeats Finder, RepeatExplorer uses relatively small genomic portions of short NGS reads, even 1% of genome coverage contains enough information about high-copy sequences in the genome. The current version of RepeatExplorer is also recommended for the development of satellite DNA probes for cytogenetic experiments and preparing pseudoshort reads from HiFi PacBio reads [39]. Additional tools within the pipeline perform automatic annotation and quantification of identified repeats. However, the default parameters of the RepeatExplorer are not suitable for analysis of low complexity sequences like microsatellites and telomeres. The Tandem Repeats Finder [234] is more suitable for such searches because it is possible to process large inputs (tens and hundreds of millions of reads) and even very short telomeres in extremely large genomes can be detected (Figure 4d). Recommended in silico workflows which can be used for any species are described in Peska et al. [206]. In the special cases of large genomes, a combination of approaches can be employed [146,204,205]. A new bioinformatic approach combining results from Tandem Repeat Finder and identification of TR subunits [132,252] can be recommended as a general approach for telomere repeat searches (Figure 4g). Experimental evaluation should then follow the identification of candidate sequences, especially when a candidate sequence is unusual for the group.

## 10. Conclusions and Perspectives

Telomere research has come a long way since the late 20th Century when telomeric sequences were first identified [76]. Many of the key techniques used in the past and which are still relevant for further research can be performed using commonplace laboratory equipment. Non-telomere satellite research does not have the same link to human health and disease and perhaps does not benefit from the same level of interest. With large amounts of genomic data now available and easily accessible, predicting or verifying laboratory experiments with in silico work is now an increasingly more sensible plan that should be accessible to any laboratory. Equally, as knowledge of telomeres and particularly subtelomeric satellites expands, it is apparent that open-access databases of this knowledge, particularly the results from karyotyping studies [151,167], would benefit the research community as a whole and make such data more visible and manageable. There are still many unanswered questions in satellite research making this an inviting topic for the open-minded researcher. Little is known about the biological significance of variant classes of ITSs, although it is clear that the phylogenic tree of life can be explored using these for simple genetic fingerprinting. Likewise, the presence or absence of satellite sequences within chromosomes and the frequent rapid changes in these is an ongoing area of research. In particular, many intriguing hypotheses have been proposed regarding horizontal genetic transfer of repetitive sequences and only few of these hypotheses have received robust experimental support [69]. Beyond simple characterization and identification, the mechanisms behind changes in the specific sequences of telomeric and other satellite repeats and the biological repercussions of these are still mysterious. In a related point, the identity of telomere maintenance and protection systems and how they have presumably evolved in parallel with changing telomeric DNA is also another relatively open topic. Most of all, it is apparent that satellite research is one area of biology where the study of just a few model organisms does not tell the complete story. Extending some of the research techniques that have been used in the past into studies of non-model organisms seems like a potential rich vein of new information. Investigating new models and keeping a sharp eye and an open mind for unusual or interesting new biological mechanisms would seem to be both a strong direction towards the future of satellite research and fully in keeping with the spirit of Mendel.

## Figures and Tables

**Figure 3 genes-13-01663-f003:**
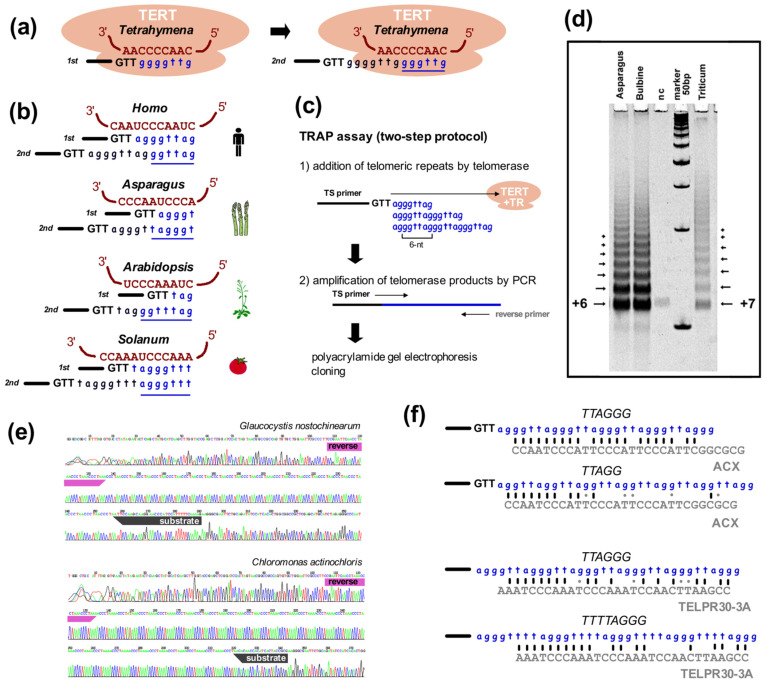
**TRAP assay helps to verify telomeric repeat synthesized by telomerase**. (**a**) Telomerase adds telomere repeats at the 3′ telomeric overhang (here shown as GTT-3′, black). Firstly, substrate ssDNA anneals to the template region of telomerase RNA (brown) and nucleotides are added (blue) up to the end of the template region. The substrate is then reannealed and a new repeat is synthesized such that one repeat (underlined) is added per round. (**b**) Representative template regions of the TR subunit. The template regions of telomerases synthesizing the same type of telomere repeat may be different, e.g., human and *Asparagus officinalis* (Asparagales), or *Arabidopsis* (Brassicales) and tomato (Solanales). (**c**) Classical TRAP is a PCR-based assay for the investigation of telomerase activity through elongation of the telomerase substrate primer, e.g., the widely used TS primer (5′-AATCCGTCGAGCAGAGTT-3′, [224,228]). The sequence added by telomerase corresponds to the template region. Elongation products are amplified by PCR with the substrate primer and a C-rich telomeric reverse primer. (**d**) A ladder of products is visualized on a polyacrylamide gel. TRAP products from Asparagales plants with a human-type telomeric repeat produced a ladder with 6-bp periodicity, while 7-bp periodicity was detected in wheat used as a control (nc—negative control) [139]. (**e**) Sequencing of cloned TRAP products revealed synthesis of the TTAGGG-type telomeric repeat in *Glaucocystis nostochinearum* (Glaucophyta, [142]) and the TTTTAGGG-type of telomeric repeat in *Chloromonas actinochloris* (Chlorophyta, [30]). Both TRAP reactions were performed using various substrate primers and a reverse primer (TELPR30-3A) corresponding to the TTTAGGG-type telomeric repeat, primer sites are highlighted. (**f**) Examples of the limited sequence similarity of reverse primers (grey) targeting telomerase elongated products. Amplification of a sequence-related type of telomeric repeat is enabled by crosshybridization of reverse primer, e.g., the ACX primer routinely used for human-type telomeric sequence was reported to detect insect telomerase activity in *Daphnia* [229] or the *Arabidopsis*-type TELPR30-3A detecting human-type or *Chlamydomonas*-type repeat (see also (**d**,**e**)). The anchor site designed at the 5′ end of reverse primers ensures preferred amplification of specific products from the first step instead of primer sliding due to multiple sequence targets in tandem repeat.

**Figure 5 genes-13-01663-f005:**
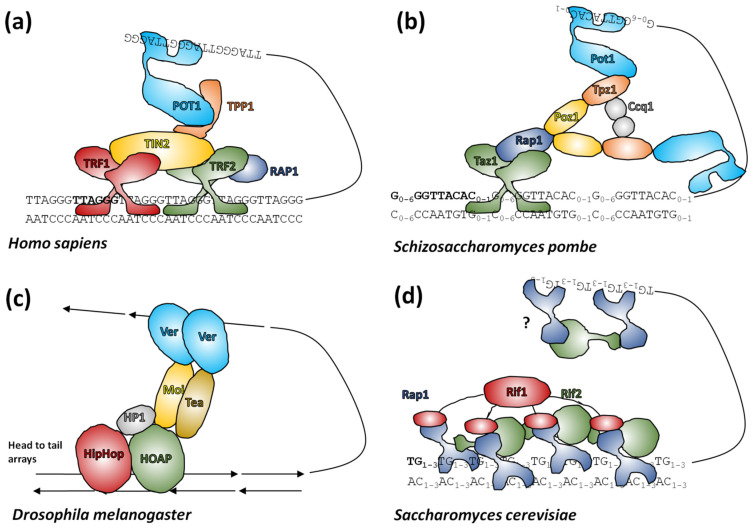
**Telomere protection by protein complexes.** (**a**) The six core units of shelterin [272] form a complex coating distal telomeres, although stoichiometries of subunits may vary. TRF2 forms T-loops and binds the double-stranded vertebrate telomeric sequence. TRF1 assists this binding, TIN2 and TPP1 form the core of the complex and control other protein-protein interactions and POT1 can bind single-stranded telomeric repeats to stabilize the T-loop. (**b**) Fission yeast shelterin [286] is analogous to vertebrates but differs in stoichiometries of proteins. (**c**) *Drosophila* terminin [95] has a similar function to shelterin although the precise roles of components that share little homology with shelterin components are speculative. (**d**) Budding yeast telosomes [287]. Rap1 binds telomeric DNA and can be complexed into dimers by Rif2 or tetramers by Rif1. The entire assembly is proposed to form a velcro-like coating of telomeres although to date structural studies of this complex are on dsDNA only, so any interaction with 3′ overhangs is speculative.

**Figure 6 genes-13-01663-f006:**
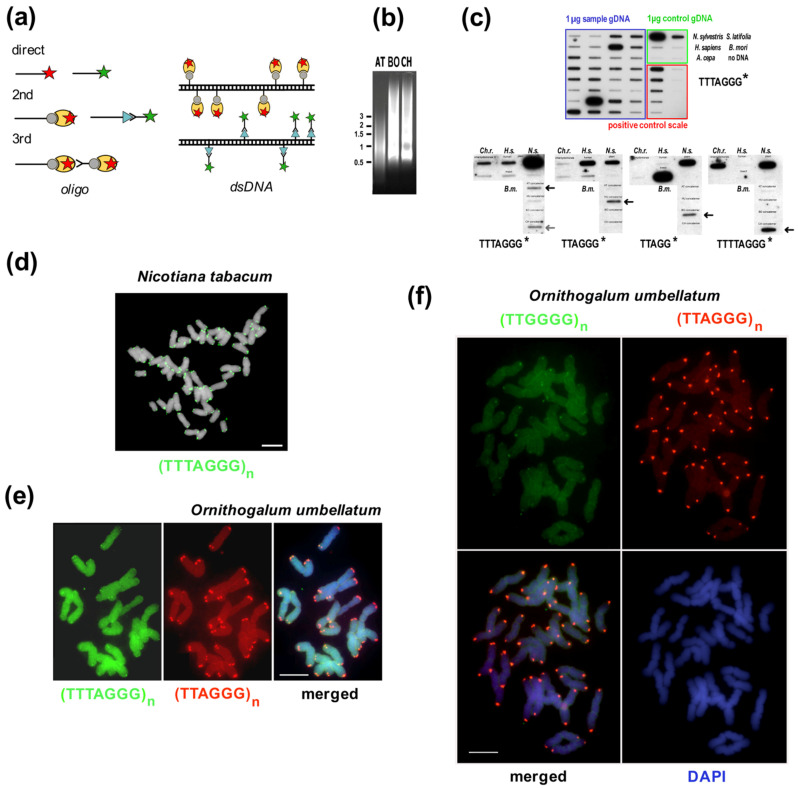
**Telomere identification by****FISH and slot-blot hybridization.** (**a**) Example FISH probes (**top** to **bottom**), custom synthesized oligonucleotides directly labeled with a fluorophore (asterisk). Biotinylated (circle) oligonucleotides, for secondary detection using streptavidin-bound fluorophore or tertiary detection, e.g., in a sandwich with biotinylated anti-streptavidin antibody and secondary antibody conjugated with a fluorophore. Digoxigenin-labeled (triangle) oligonucleotides are visualized by an anti-digoxigenin antibodies conjugated with a fluorophore. dsDNA probes for FISH and other hybridization techniques are often labeled in PCR reactions or by nick translation using biotin/digoxigenin modified nucleotides incorporated into products. (**b**) Template-free PCR produces a typical smear of products generated by self-annealing of corresponding G- and C- telomeric oligonucleotides (AT, *Arabidopsis*-type TTTAGGG/CCCTAAA; BO, *Bombyx*-type TTAGG/CCTAA; CH, *Chlamydomonas*-type TTTTAGGG/CCCTAAAA, [198,322]) and amplification by Taq DNA polymerase. Concatemers can be labeled radioactively for Southern hybridization or can have modified nucleotides incorporated as for FISH. (**c**) Slot-blot hybridization screening for telomere sequence signals using radioactively (*) labeled oligonucleotide probes. (**Upper** panel) gDNA samples of interest (24 species, blue frame, modified from [139]) and controls are immobilized on membrane. Control gDNA samples (green frame) and serial dilutions of control plasmids with cloned telomere sequences (red frame) serve as standards for normalization of signals between membranes when larger collections are investigated. N.B. *N. sylvestris* and *S. latifolia* are of similar genome size but differ significantly in length of telomeres. (**Lower** panels) Slot-blot hybridization of gDNA sets from representative species (N.s., *N. sylvestris*; H.s., *Homo sapiens*; Ch.r., *Chlamydomonas reinhardtii*; B.m., *Bombyx mori*) using four telomere probes, 100 pg of the respective concatemers (arrows) serve as a control for mutual comparison. Note crosshybridization of the oligonucleotide probe TTTAGGG to concatemers of the *Chlamydomonas* telomere type (grey arrow) [322]. (**d**) FISH of *Nicotiana tabacum* metaphase chromosomes using (TTTAGGG)_n_ concatemers (green) clearly marked all termini (scale bar is 10 µm, [158]; modified). (**e**,**f**) A mixture of telomere variants occur in telomeres of *Ornithogalum umbellatum*. Scale bars are 5 µm. (**e**) The *Arabidopsis*-type telomeric probe (green) shows signals at some chromosome termini at incomplete metaphase of *O. umbellatum* and these overlap with human-type telomeric probe (red) which is dominant and represents the true telomere repeat. (**f**) Metaphase *O. umbellatum* chromosomes labeled with *Tetrahymena*-type concatemers ((TTGGGG)_n_, green) and image merged with labeled human-type concatemers (red) showing similar results to (**e**) ([139]; modified). All three variants of telomeric repeats occur in the *Ornithogalum* genome, thus FISH signals of the *Arabidopsis*-type (**e**) and *Tetrahymena*-type (**f**) variants do not result from crosshybridization and the human type was proven as the true telomere [139]. Pictures are adapted, courtesy of Dr. Terezie Mandáková (**d**) and Prof. Andrew Leitch (**e**,**f**), chromosomes were counterstained with DAPI.

## Data Availability

All data are described and referred in this paper.

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
