# Peer review of "Telomeres and Their Neighbors"

_genes, 2022, doi:10.3390/genes13091663_

Round 1
Reviewer 1 Report
In the contest of the celebration of Mendel’s anniversary, at 200 years after his birth, the authors propose this review with the aim to describe how accessible experimental and in silico methods contributed to fundamental advancements in the telomere biology field, illustrating the principals behind experiments. The review is well organized and written, mainly focusing on plants but also extending to yeast, insects and human. After a beginning part which is a little didactic, the review seems to be complete, touching a number of techniques with related applications and examples.
Author Response
Thank you very much for your kind words. In the revised version, we have accepted changes recommended by all reviewers, corrected some mistyping, completed references with doi numbers and clarified better the focus of our review in the abstract. We hope reviewer #1 will be satisfied with changes and the revised manuscript can be accepted for publication.
Reviewer 2 Report
Telomeres, chromosome instability and cancer by Susan M. Bailey and John P. Murnane is very complete, well written and readable by a relatively broad audience.
That said, I have a few suggestions that came to mind after reading through multiple times.
There is a bit of disconnect between the abstract and the text. Some of the items listed in the second sentence of the abstract are barely touched upon in the text, while other items covered at length in the text are not mentioned in the abstract. Please consider deleting the second sentence of the abstract, it directs the readers attention to areas that are not the strong points in the text.
The disconnect appears to stem from a change in focus. The text focuses on non-model organisms. This focus is a good, because there are many reviews of telomeres in model organisms and these reviews are properly cited, but not otherwise discussed.
An area poorly covered in the text is the B/F/B cycle and chromosomal instability. The B/F/B cycle is mentioned only once in the text, and sister chromatid pairing, non-homologous end-joining, and other chromosomal change is glossed over and not mentioned. The second sentence of the introduction uses an interesting term, “attrition”. The paper would be stronger and appeal to a broader audience by addition of a paragraph or two that focuses directly on that attrition.
Other than the above, the review is outstanding. The figures are extensive, but readable and relatively easy to follow. The authors unique area of expertise is reflected well throughout. Their argument in favor of studies with non-model organisms is compelling.
The tie-in with Mendel is minimal, but as stated, is “in the spirit of Mendel.”
Author Response
Thank you very much for your kind words and your time for thoroughly reviewing the text. Based on your comments we have clarified better the focus of the review in the abstract. We agree that we had glossed over chromosomal rearrangement in the original manuscript and this would be something well worth including, especially if this points new researchers towards an exciting and open field. As such we have added in a paragraph about B/F/B cycles and chromosome instability along with mentions of this in the abstract and conclusions. Moreover, we have accepted changes recommended by all reviewers, corrected some mistyping and completed references with doi numbers. We hope the reviewer #2 will be satisfied with changes and the revised manuscript can be accepted for publication.
Reviewer 3 Report
Dear authors, the paper is very interesting and I have only a minor suggestion. To better describe the knowledge about satellite DNA evolution you should mentioned the molecular mechanism of satellite DNA evolution based as it was described for European brown frog species.
Author Response
Thank you very much for your kind words and to pointing us towards an interesting area of research that we had omitted. In the revised version, we have added a paragraph on the molecular mechanisms of satellite DNA evolution as exemplified by studies on European brown frogs and reflected this area of research in the abstract and conclusions. Moreover, we have accepted changes recommended by all reviewers, corrected some mistyping, completed references with doi numbers and clarified better focus of our review in the abstract. We hope the reviewer #3 will be satisfied with changes and the revised manuscript can be accepted for publication.
Reviewer 4 Report
The review by Jenner et al. provides an extensive, nice, rich, and detailed overview about an interesting topic as telomeres. They give from a general introduction to a more specific information in the knowledge of telomeres and satellite systems, including a particular and helpful focus on accessible experimental and in silico methods for the study of telomeres. They also include a considerable number of references of articles and reviews highly valuable for the interested reader. It is important to highlight that text is also supported with clear figures where needed.
I have only minor suggestions to improve the manuscript:
1. Line 26-27. “Eukaryotic chromosomes are formed from key DNA structures: centromeres, telomeres and origins of replication”. Please consider modification to a more precise sentence.
2. Line 43. Please review “drosophila”.
3. Line 503. Please review “in silico”.
4. Figure 5b. Please review a missing “s” in Schizosaccaromyces pombe.
Author Response
Thank you very much for your kind words and suggestions.
Following your recommendations,
- we modified the first sentence of introduction. Now it reads „The essential DNA structures that form eukaryotic chromosomes are centromeres, telomeres and origins of replication.“
- Line 43. „drosophila“ was corrected to „Drosophila“,
- Line 503. „in silico“ was unified to italics and corrected also in other places within manuscript,
- mistyping in Figure 5b was corrected.
Moreover in the revised version, we have accepted changes recommended by all reviewers including additional paragraphs, we completed references with doi numbers and clarified better focus of our review in the abstract. We hope reviewer #4 will be satisfied with changes and the revised manuscript can be accepted for publication.